

# Genome-wide analysis of *BpDof* genes and the tolerance to drought stress in birch (*Betula platyphylla*)

Shilin Sun[1,2,*], Bo Wang[3,*], Qi Jiang[3], Zhuoran Li[1,2], Site Jia[1,2], Yucheng Wang[1,2] and Huiyan Guo[1,2]

[1] College of Forestry, Shenyang Agricultural University, Shenyang, Liaoning, China
[2] The Key Laboratory of Forest Tree Genetics, Breeding and Cultivation of Liaoning Province, Shenyang Agricultural University, Shenyang, Liaoning, China
[3] Department of Life Science and Technology, Mudanjiang Normal University, Mudanjiang, Heilongjiang, China
[*] These authors contributed equally to this work.

## ABSTRACT

**Background**. DNA binding with one finger (Dof) proteins are plant-specific transcription factors playing vital roles in developmental processes and stress responses in plants. Nevertheless, the characterizations, expression patterns, and functions of the Dof family under drought stress (a key determinant of plant physiology and metabolic homeostasis) in woody plants remain unclear.

**Methods**. The birch (*Betula platyphylla var. mandshuric*) genome and plant TFDB database were used to identify *Dof* gene family members in birch plants. ClustalW2 of BioEdit v7.2.1, MEGA v7.0, ExPASy ProtParam tool, Subloc, TMHMM v2.0, GSDS v2.0, MEME, TBtools, KaKs Calculator v2.0, and PlantCARE were respectively used to align the *BpDof* sequences, build a phylogenetic tree, identify the physicochemical properties, analyze the chromosomal distribution and synteny, and identify the *cis*-elements in the promoter regions of the 26 *BpDof* genes. Additionally, the birch seedlings were exposed to PEG6000-simulated drought stress, and the expression patterns of the *BpDof* genes in different tissues were analyzed by qRT-PCR. The histochemical staining and the evaluation of physiological indexes were performed to assess the plant tolerance to drought with transient overexpression of *BpDof4*, *BpDof11*, and *BpDof17* genes. SPSS software and ANOVA were used to conduct all statistical analyses and determine statistically significant differences between results.

**Results**. A total of 26 *BpDof* genes were identified in birch via whole-genome analysis. The conserved Dof domain with a C(x)2C(x)21C(x)2C zinc finger motif was present in all BpDof proteins. These birch *BpDofs* were classified into four groups (A to D) according to the phylogenetic analysis of *Arabidopsis thaliana Dof* genes. BpDof proteins within the same group mostly possessed similar motifs, as detected by conserved motif analysis. The exon–intron analysis revealed that the structures of *BpDof* genes differed, indicating probable gene gain and lose during the *BpDof* evolution. The chromosomal distribution and synteny analysis showed that the 26 *BpDofs* were unevenly distributed on 14 chromosomes, and seven duplication events among six chromosomes were found. *Cis*-acting elements were abundant in the promoter regions of the 26 *BpDof* genes. qRT-PCR revealed that the expression of the 26 *BpDof* genes was differentially regulated by drought stress among roots, stems, and leaves. Most

Corresponding author
Huiyan Guo, lxyghy@syau.edu.cn

*BpDof* genes responded to drought stress, and *BpDof4*, *BpDof11*, and *BpDof17* were significantly up-regulated. Therefore, plants overexpressing these three genes were generated to investigate drought stress tolerance. The *BpDof4*-, *BpDof11*-, and *BpDof17*-overexpressing plants showed promoted reactive oxygen species (ROS) scavenging capabilities and less severe cell damage, suggesting that they conferred enhanced drought tolerance in birch. This study provided an in-depth insight into the structure, evolution, expression, and function of the *Dof* gene family in plants.

# INTRODUCTION

Transcriptional regulation of gene expression in plants has a vital role in controlling or influencing many critical biological processes, such as cellular morphogenesis, signal transduction, and adverse environmental stress responses (*Borrego-Benjumea et al., 2020*; *Riechmann & Ratcliffe, 2000*; *Yang et al., 2020*). Transcription factors (TFs) can control the expression of genes by binding to specific cis-elements in their promoter regions and activating or repressing the transcription of the target genes (*Wang et al., 2015a*). Multitudinous TF families have been found in plants, including NAM/ATAF1/CUC2 (NAC), basic leucine zipper (bZIP), APETALA2/ethylene-responsive element (ERE)–binding factor (AP2/ERF), basic helix-loop-helix (bHLH), myeloblastosis (MYB), DNA binding with one finger (Dof), and Cys2 (C2) His2-type zinc fingers (*Yamasaki et al., 2013*).

The Dof proteins are a family of plant-specific TFs containing a highly conserved 50- to 56-amino acid Dof DNA-binding domain in the N-terminal region (*Gupta et al., 2015*). The Dof domain is structured as a Cys2/Cys2 $Zn^{2+}$ finger structure that regulates both DNA–protein and protein–protein interactions (*Yanagisawa, 2002*), and recognizes the specific cis-element of (AT)/AAAG in their target gene promoter region, except for a pumpkin Dof protein that binds to an AGTA motif (*Diaz et al., 2002*; *Yanagisawa, 2002*). The C-terminal of Dof proteins contains a transcriptional regulation domain that can interact with various regulatory proteins and activate the expression of the target genes (*Ma et al., 2015*).

Dof TFs are involved in many plant-specific physiological processes, including light responsiveness, seed maturation or germination, tissue differentiation, phytochrome and metabolic regulation (*Cheng et al., 2018*; *Gupta et al., 2015*; *Noguero et al., 2013*; *Zang et al., 2017*). For example, a previous study showed that the Dof protein DOF AFFECTING GERMINATION (DAG2) was a positive regulator of the light-mediated seed germination process in *Arabidopsis* (*Santopolo et al., 2015*). The RNA-seq analysis indicated that the *DAG1* promoted hypocotyl elongation *via* regulating the ABA, ethylene, and auxin signals (*Lorrai et al., 2018*). The overexpression lines of *SCAP1* (a Dof transcription factor) increased the number of guard cells (GCs) and protodermal cells recruited in the GC lineage and altered GC distribution and spacing patterns, indicating that *SCAP1* could

integrate different aspects of GC biology, including specification, spacing, and maturation (*Castorina et al., 2016*). *SlDof10* regulated the vascular tissue formation during ovary development in tomatoes (*Rojas-Gracia et al., 2019*). The overexpression of *AtDOF5.4/OBP4* in *Arabidopsis* reduced the cell size and number and resulted in dwarf plants, which strongly suggested that *OBP4* was a negative regulator of cell cycle progression and cell growth (*Xu et al., 2016b*). A *PpDof5* transcription factor in *Pinus pinaster* played a vital role in controlling ammonium assimilation for glutamine biosynthesis in conifers (*Rueda-Lopez et al., 2008*). Furthermore, the overexpression of *PpDof5* exhibited higher growth in transgenic hybrid poplars than in controls, enhanced the capacity for inorganic nitrogen uptake, and caused significantly increased accumulation of carbohydrates (*Rueda-Lopez et al., 2017*).

Some studies showed Dof TFs play important roles in response to biotic and abiotic stresses in plants (*Zang et al., 2017*). The gain- or loss-of-function analysis of *SlDof22* in tomatoes showed that *SlDof22* affected ascorbate accumulation and enhanced the tolerance to salinity in plants (*Cai et al., 2016*). The *GhDof1* of *Gossypium hirsutum* improved salt and cold tolerance and seed oil content in transgenic cotton (*Su et al., 2017*). The *OsDof15*-mediated ethylene biosynthesis played an important role in inhibiting primary root elongation by salt stress in rice (*Qin et al., 2019*). *MdDof54*-overexpressing plants had higher photosynthesis rates and shoot hydraulic conductivity under long-term drought stress and higher survival percentages under short-term drought stress compared with nontransgenic plants, illustrating that *MdDof54* could improve the drought resistance (*Chen et al., 2020b*).

Although *Dofs* have been investigated in diverse biological processes, their functional roles and regulatory mechanisms remain unclear. In addition, many members of the Dof family have not yet been characterized, especially in woody plants. Birch (*Betula platyphylla*) is a valuable broad-leaved pioneer tree of eastern Asia. It is important to stabilize forest ecosystems and regeneration. Birch is also widely used in architecture, furniture, and paper production (*Kitao et al., 2001*; *Pamidi et al., 2020*; *Xing & Liu, 2012*). In this study, 26 *Dof* genes were isolated from birch and identified to characterize the sequence and expression patterns of the birch *Dof* genes, followed by multiple sequence alignment, phylogenetic analysis, conserved motif identification, gene structure characterization, and cis-regulatory element analysis. The expression patterns of the 26 birch *Dof* genes were analyzed in roots, stems, and leaves at different times under drought stress. The *Dof* genes significantly up-regulated under drought stress were selected to examine the drought stress tolerance in transgenic plants. The results of this study will provide useful information for further investigation of the functional and regulatory mechanisms of these *Dof* genes in resistance to abiotic stress in birch.

# MATERIALS & METHODS

## Identification of *Dof* genes from *B. platphylla*

The assembled birch genome (*Chen et al., 2021*) was analyzed, and the unigenes were searched using BLASTX against the NR and Swiss-Prot databases for functional

annotation (*Camacho et al., 2009*). All BpDofs (Table S1) were detected from the birch genome by employing a hidden Markov model (HMM) profile of the Dof domain (PF02701) obtained from Pfam (http://pfam.xfam.org) using the HMMER3.0 program (http://hmmer.janelia.org) (*Finn et al., 2016*). The conserved domains of these putative BpDof proteins were identified by searching against the NCBI's conserved domain database (https://www.ncbi.nlm.nih.gov/Structure/cdd/wrpsb.cgi?) (*Marchler-Bauer et al., 2015*). Predictions of theoretical molecular size, isoelectric point (pI), and other physicochemical properties were conducted using the ExPASy ProtParam tool (http://www.expasy.org/tools/protparam.html) (*Gasteiger et al., 2003*). Nuclear localization signals and transmembrane domains were predicted using SubLoc (http://cello.life.nctu.edu.tw/cello2go/) and TMHMM server 2.0 (http://www.cbs.dtu.dk/services/TMHMM/), respectively (*Chen, Huang & Sun, 2006*; *Krogh et al., 2001*).

## Sequence alignment and phylogenetic analysis of the 26 Dof proteins

The multiple sequence alignments of 26 birch Dof proteins were arrayed in the ClustalW2 of BioEdit7.2.1 software (http://www.ebi.ac.uk/Tools/clustalw2/) (*Larkin et al., 2007*). A phylogenetic tree of 26 birch Dof proteins with the 39 Dof proteins of *Arabidopsis* (Table S2) was constructed using MEGA7.0 by the neighbor-joining (NJ) method (*Kumar, Stecher & Tamura, 2016*; *Tamura et al., 2011*). The phylogenetic relationships of 26 birch Dof proteins were analyzed.

## Gene structure and conserved motif analysis

The genome sequences of the 26 *Dof* genes were acquired from the birch genome, and their exon/intron structures were analyzed using the GENE Structure Display Serve 2.0 (http://gsds.cbi.pku.edu.cn/) (*Hu et al., 2015*). Their conserved motifs with default parameters were analyzed using MEME (http://meme-suite.org/tools/meme) (*Bailey et al., 2009*), but the maximum number of motifs was set as 15.

## Chromosomal distribution and synteny analysis

The length of each chromosome and the location of each *BpDof* gene (Table 1) were retrieved from the birch genome, and the chromosomal distribution of *BpDof* genes was visualized with the Amazing Super Circos software in TBtools (*Chen et al., 2020a*). One-step MCScanX SuperFast software in TBtools was used for gene synteny and collinearity analyses with default parameters, and the syntenic map was constructed with the chromosomal locations of *BpDofs*. Furthermore, to analyze the selection pressure of *BpDofs*, the non-synonymous rate (Ka), synonymous rate (Ks), and Ka/Ks values of the corresponding *BpDofs* were calculated using KaKs Calculator v2.0 (*Wang et al., 2009*).

## Cis-element analysis in the promoters of 26 *BpDofs*

The 2,000-bp length of the upstream DNA sequence with the 5′-untranslated region (UTR) for each *Dof* gene was obtained from the birch genome. The cis-elements in the promoter sequences of the 26 *BpDof* genes were analyzed using the PlantCARE database (http://bioinformatics.psb.ugent.be/webtools/plantcare/html/) (*Lescot et al., 2002*).

**Table 1  Characterization of 26 *BpDof* transcription factors.**

| Name | Locus ID | Chromosome position | Len | MW | PI | AI | II | Stability | GRAVY | CR |
|---|---|---|---|---|---|---|---|---|---|---|
| BpDof1 | BPChr06G16490 | Chr06:3395833:3404094 | 310 | 33.2 | 9.03 | 61.68 | 56.67 | Unstable | −0.525 | + |
| BpDof2 | BPChr12G11401 | Chr12:1468437:1469132 | 231 | 24 | 6.06 | 49.83 | 54.00 | Unstable | −0.436 | − |
| BpDof3 | BPChr11G05806 | Chr11:4202830:4206047 | 562 | 61 | 5.15 | 53.72 | 58.45 | Unstable | −0.79 | − |
| BpDof4 | BPChr12G08354 | Chr12:5473320:5474114 | 264 | 29.4 | 4.97 | 59.47 | 66.08 | Unstable | −0.684 | − |
| BpDof5 | BPChr14G09159 | Chr14:8257436:8258170 | 244 | 27.1 | 6.24 | 55.9 | 54.33 | Unstable | −0.595 | − |
| BpDof6 | BPChr06G09621 | Chr06:36382710:36383854 | 326 | 35.3 | 8.89 | 48.47 | 55.66 | Unstable | −0.771 | + |
| BpDof7 | BPChr14G12625 | Chr14:5137993:5139183 | 340 | 35.6 | 9.36 | 52.06 | 53.49 | Unstable | −0.549 | + |
| BpDof8 | BPChr12G29175 | Chr12:10562079:10563017 | 312 | 33.9 | 7.18 | 57.24 | 57.26 | Unstable | −0.678 | = |
| BpDof9 | BPChr06G29469 | Chr06:6207612:6212636 | 265 | 29.1 | 9.13 | 46.38 | 41.85 | Unstable | −0.829 | + |
| BpDof10 | BPChr02G19918 | Chr02:22002669:22010046 | 248 | 27.3 | 9.07 | 57.38 | 49.41 | Unstable | −0.676 | + |
| BpDof11 | BPChr10G04282 | Chr10:1350584:1369701 | 295 | 32.8 | 7.61 | 48.27 | 49.33 | Unstable | −0.886 | + |
| BpDof12 | BPChr06G19208 | Chr06:11028826:11029461 | 220 | 23.8 | 8.53 | 43.05 | 49.18 | Unstable | −0.794 | + |
| BpDof13 | BPChr03G28866 | Chr03:20524755:20525537 | 260 | 29.3 | 8.44 | 50.65 | 72.06 | Unstable | −0.968 | + |
| BpDof14 | BPChr04G00494 | Chr04:8288764:8291483 | 517 | 56.8 | 7.12 | 62.44 | 52.71 | Unstable | −0.625 | = |
| BpDof15 | BPChr04G23864 | Chr04:3304076:3305071 | 331 | 35.4 | 9.12 | 53.66 | 46.02 | Unstable | −0.644 | + |
| BpDof16 | BPChr06G02126 | Chr06:1426392:1427096 | 234 | 24.1 | 8.51 | 50.9 | 37.07 | Stable | −0.438 | + |
| BpDof17 | BPChr07G09798 | Chr07:24560688:24561170 | 160 | 17.8 | 9.01 | 49.38 | 43.05 | Unstable | −0.803 | + |
| BpDof18 | BPChr07G18918 | Chr07:19263129:19264016 | 295 | 32.7 | 9.15 | 40.34 | 66.71 | Unstable | −1.082 | + |
| BpDof19 | BPChr07G18939 | Chr07:19281783:19282965 | 342 | 37.5 | 8.43 | 52.19 | 53.51 | Unstable | −0.758 | + |
| BpDof20 | BPChr08G01518 | Chr08:8408149:8408980 | 182 | 20.6 | 8.99 | 51.48 | 33.69 | Stable | −0.866 | + |
| BpDof21 | BPChr08G17028 | Chr08:39507804:39510785 | 490 | 53.3 | 6.71 | 57.14 | 59.52 | Unstable | −0.773 | − |
| BpDof22 | BPChr11G09292 | Chr11:23999498:24000624 | 319 | 34.5 | 9.21 | 59.97 | 66.82 | Unstable | −0.643 | + |
| BpDof23 | BPChr11G10185 | Chr11:35669848:35671309 | 297 | 32.9 | 8.54 | 49.26 | 53.21 | Unstable | −0.795 | + |
| BpDof24 | BPChr12G29204 | Chr12:9026101:9027218 | 249 | 27 | 9.32 | 47.43 | 50.63 | Unstable | −0.738 | + |
| BpDof25 | BPChr13G02551 | Chr13:5309999:5313177 | 510 | 55.6 | 6.54 | 52.43 | 57.19 | Unstable | −0.731 | − |
| BpDof26 | BPChr14G05515 | Chr14:9880786:9891340 | 518 | 57.1 | 6.19 | 61.39 | 52.63 | Unstable | −0.554 | − |

Notes.

AI, Aliphatic index; CR, charged residues (positively: +; negatively: ; neutral: =); GRAVY, grand average of hydropathicity; II, instability index; MW, protein molecular size (kDa); PI, isoelectric point.

## Plant materials and drought stress treatment

Birch seeds were placed in a glass bottle loosely wrapped with gauze and rinsed with water for 3 days. After filtering out most of the water, the seeds were evenly spread on the soil (soil:vermiculite:perlite = 3:1:1). A thin layer of soil was applied to cover the seeds, after which they were watered and sealed using preservative film with some holes in it. The film was removed from the seeds after germination. Once the seedlings grew to 3–4 cm, the seedlings with uniform growth and in good conditions were selected. One birch seedling was planted in one pot. After about 2 months cultivation, the healthy birch seedlings about 25 cm height with similar growth conditions were treated using 20% PEG6000 for 0.5, 1, 3, 6, 12, and 24 h in a reversed time order. The control plants were treated with fresh water for 24 h. Each birch seedling was watered with 1 L of 20% PEG6000 or water. The developing roots, stems, and leaves of the birch seedlings under drought treatments were

collected after 24 h. All samples were immediately frozen in liquid nitrogen and stored at −80 °C. Three biological replicates were conducted in each experiment.

## RNA isolation and real-time PCR validation

The RNA of each sample was extracted using a Universal Plant RNA Extraction Kit (BioTeke Corporation, China) from 100 mg plant tissues (roots, stems, or leaves) of birch plants, and cDNA was synthesized from approximately 1 μg total RNA using PrimeScript IV First-Strand cDNA Synthesis Mix (TaKaRa, Japan). A 20- μL volume containing 10 μL of SYBR Green Real-time PCR Master Mix (BioTake Corporation, China), 1 μL of cDNA template, and 1 μL each of the forward primer (10 μM) and reverse primer (10 μM) was used (all primers and amplicon sizes are shown in Table S3), after which ultrapure $H_2O$ was used to make up the reaction volume. The amplicons were completed as follows: 94 °C for 30 s; 94 °C for 12 s, 58 °C for 30 s, and 72 °C for 45 s, for 45 cycles, followed by 79 °C for 1 s for plate reading using an qTOWER$^3$ G, analytik Jena AG, Germany. After the final PCR cycle, the temperature of 0.5 °C per second was increased from 55 °C to 99 °C to generate the melting curve for samples. The relative mRNA levels were determined by normalizing the PCR threshold cycle number of each gene to that of ubiquitin (GenBank number: FG065618) and α-tubulin (GenBank number: FG067376) as internal references for all treatments. The threshold for the Ct values was the machine setting, and the average Ct value was calculated using three biological replicates. The relative expression levels of the 26 *BpDof* genes were calculated from the threshold cycle by the delta–delta CT method (*Pfaffl, Horgan & Dempfle, 2002*).

## Vector construction and transient transformation

The full-length coding sequences (CDSs) of *BpDof4*, *BpDof11*, and *BpDof17* were amplified by PCR,and then constructed into the pROKII vector digested with *Sma* I (NEB, USA) using the In-Fusion$^{TM}$ CF Liquid PCR Cloning kit (Takara, Japan) under control of the CaMV 35S promoter for overexpression of *BpDof4*, *BpDof11*, and *BpDof17* (35S:*BpDof* ), respectively. The primers used for PCR are shown in Supplemental Table S4. The pROKII-35S::*BpDof4*, 35S::*BpDof11*, and 35S::*BpDof17* were separately transformed into 4-week-old birch seedlings by *Agrobacterium tumefaciens*–mediated transient expression (*Zhang, Wang & Wang, 2012*) with some modifications. In brief, Luria-Bertani (LB) liquid medium supplied with 50 mg/L kanamycin and 50 mg/L rifampicin was used to culture the *A. tumefaciens* strain EHA105 transformed with pROKII-35S::*BpDof4*, pROKII-35S::*BpDof11*, pROKII-35S::*BpDof17*, or the empty pROKII-35S vector. *A. tumefaciens* cultures were re-suspended in the transformation solution (1/2 MS + sucrose [2.0%, w/v] + 10 mM $CaCl_2$ + 120 μM acetosyringone + 200 mg/L DTT + Tween-20 [0.02%, v/v], pH 5.8], which were then harvested at an $OD_{600}$ of 0.6 by centrifugating at 3000 g for 10 min. For transient genetic transformation, the plants were soaked into the transformation solution and shaken at 120 rpm and 25 °C for 2 h. Then the plants were planted vertically on 1/2 MS solid medium (1/2 MS + sucrose [2.0%, w/v] + 120 μM acetosyringone + 200 mg/L DTT, pH 5.8) and incubated at 25 °C in the dark. After culturing for 48 h, the plants were assumed to have been transformed and were then used for subsequent experiments.

### Stress tolerance analysis of *BpDof4-, BpDof11-,* and *BpDof17-overexpressing plants*

The birch plants overexpressing *BpDof4*, *BpDof11*, or *BpDof17* were treated with 20% PEG6000 for 6 h. The pROKII-35S transformants and the wild-type (WT) birch seedlings were also treated with PEG6000. Water treatment was used as control. The detached leaves of birch plants were incubated with 0.5 mg/mL nitroblue tetrazolium (NBT, dissolved in phosphate buffer, pH 7.8) and 1.0 mg/mL 3′-diaminobenzidine (DAB, dissolved in phosphate buffer, pH 3.8) as described previously (*Zhang et al., 2011*). Evans blue (1.0 mg/mL, dissolved in sterile deionize water) staining was performed to detect cell death following the published protocols (*Kim et al., 2003*). Superoxide dismutase (SOD) and peroxidase (POD) activities, $H_2O_2$ content, and electrolyte leakage were measured as previously described (*Liu et al., 2015*; *Wang et al., 2015b*). Three independent biological replicates were performed.

### Statistical analysis

All statistical analyses were performed using SPSS software (IBM, IL, USA), and ANOVA was used to determine statistically significant differences between results. The level of significance was set at $P < 0.05$.

## RESULTS

### Identification and characterization of the 26 *Dofs* in *B. platyphylla*

Twenty-six full-length Dof TFs (GenBank accession numbers: MW538484–MW538509) were identified in *B. platyphylla* (Table 1) using the HMMER3.0 program with a HMM profile of the Dof domain (PF02701), and conserved domains of these putative BpDof proteins were identified by searching against the NCBI's conserved domain database. These proteins encoded by *BpDof* genes consisted of 160–562 amino acids (aa). The molecular sizes and pI values of these proteins ranged from 17.8 kDa to 61.0 kDa and 4.97 to 9.36, respectively, and their aliphatic indexes were between 43.34 and 62.44. Analyses of instability indexes and grand average of hydropathicities indicated that most BpDof proteins were unstable hydrophilic proteins except BpDof16 and BpDof20. The charge results showed that BpDof8 and BpDof14 were neutral; BpDof2, BpDof3, BpDof4, BpDof5, BpDof21, BpDof25, and BpDof26 were negative; and the other 17 BpDofs were positive. All BpDof proteins were predicted to be localized to the nucleus. The transmembrane domain analysis indicated that the 26 BpDof proteins did not have α-helical transmembrane motifs (Tables S5 and S6).

### Sequence alignment and phylogenetic analysis of BpDof proteins

The multiple sequence alignments of the 26 birch BpDof proteins (Table S1), together with several representative Dof proteins selected from the published *Arabidopsis* databases (https://www.arabidopsis.org) and *Populus trichocarpa v3.0* genomics resource (https://phytozome.jgi.doe.gov/pz/portal.html#!info?alias=Org_Ptrichocarpa) (Table S7), were analyzed. The single Dof domain with the C(x)2C(x)21C(x)2C zinc finger pattern was harbored in the N-terminal region of all the putative 26 *BpDofs* (Fig. 1).
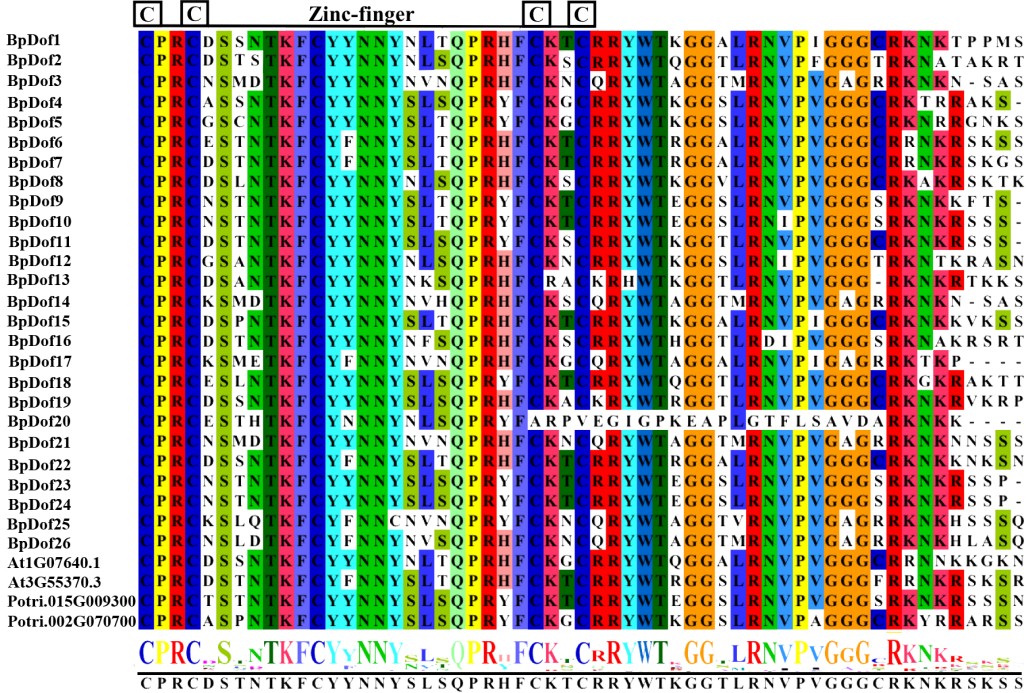

**Figure 1** **Identification of conserved Dof subdomains in 26 BpDofs.** Multiple sequence alignments of 26 BpDofs from birch with two AtDofs from Arabidopsis and 2 PtDofs from popular were obtained using ClustalW in the BioEdit software. The zinc finger structure was displayed under multiple sequence alignment results, and the font size in the zinc finger structure represented the frequency of the respective amino acid.

The phylogenetic relationships of the 26 BpDof protein sequences, along with the AtDof protein sequences in *A. thaliana*, were examined using the NJ method. Results showed that the 26 BpDof proteins were classified into four major groups (A to D) (Fig. 2). The subfamily D1 in Group D was the largest group, which included seven members accounting for 26.9% of all BpDof proteins. The subfamily B1 consisted of five members, accounting for 19.2%. The subfamily C2.1 comprised four members, accounting for 15.4%. Group A and subfamilies C1, C2.2, and D2 contained two members accounting for 7.7%, respectively. Subfamilies B2 and C3 with only one member had a proportion of 3.8%, respectively.

## Conserved motifs and gene structure analysis

Conserved motifs were identified using the MEME tool, and an unrooted phylogenetic tree was constructed based on BpDof protein sequences (Fig. 3A). Different numbers of conserved motifs were set in MEME so as to find the most significant conserved motifs in the 26 BpDof proteins based on the statistical significance ($E$ value $< 10^{-5}$). The results indicated a total of 15 conserved motifs, and the 26 BpDof proteins were classified into four main groups (A–D) including nine subfamilies basing on the phylogenetic tree. Among them, motif 1 and motif 2 were common motifs shared in almost all BpDof proteins except that BpDof20 lacked motif 2, implying that they were conserved motifs. Some of the BpDof proteins possessed specific motifs, for example, motif 7 was only present in

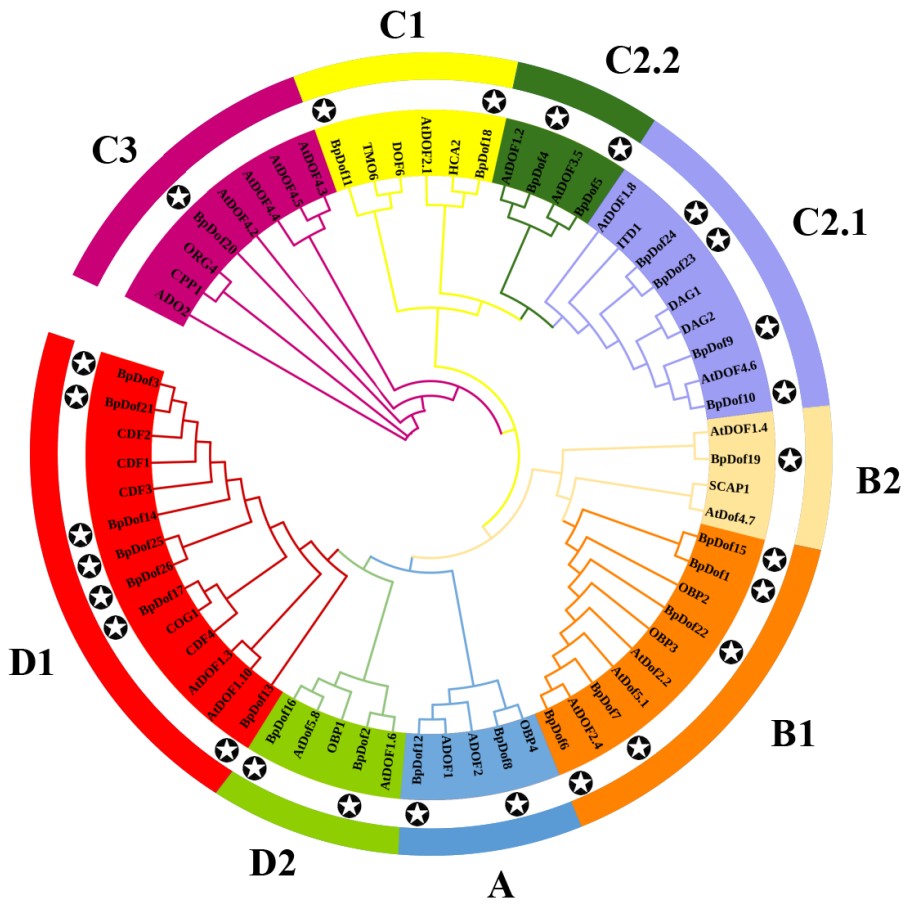

**Figure 2 Phylogenetic analysis of 26 BpDof protein sequences in birch with AtDof protein sequences in Arabidopsis.** The phylogenetic analysis of 26 BpDof protein sequences in birch and 39 AtDofs protein sequences in Arabidopsis was performed using MEGA 7.0. The full-length amino acid sequences of all Dof proteins were aligned using ClustalX 1.83. The star in the black circle represents *BpDofs*.

BpDof3, BpDof14, BpDof17, BpDof21, and BpDof26, and motif 11 and motif 13 were only distributed in BpDof6, BpDof7, and BpDof22, suggesting these motifs may be relevant to various functions of *BpDof* genes.

The transcript sequences of 26 *BpDofs* were compared with genomic sequences to obtain the distribution of introns and exons (Fig. 3B). The number of introns in the 26 *BpDofs* ranged from 0 to 2. As a result, 12 *BpDof* genes contained only exons without any introns (*BpDof1*, *BpDof2*, *BpDof4*, *BpDof5*, *BpDof8*, *BpDof11*, *BpDof12*, *BpDof13*, *BpDof15-BpDof18*), 9 *BpDof* genes (*BpDof6*, *BpDof7*, *BpDof14*, *BpDof19-BpDof22*, *BpDof24* and *BpDof25*) respectively contained one intron and two exons, whereas 5 *BpDof* genes (*BpDof3*, *BpDof9*, *BpDof10*, *BpDof23* and *BpDof26*) contained two introns and three exons.

## Chromosomal distribution and inter-specific synteny analysis of *BpDofs*

We mapped the birch *Dof* family genes on birch chromosomes to obtain their chromosomal distribution. The results (Fig. 4) showed that the 26 *BpDofs* were unevenly distributed on

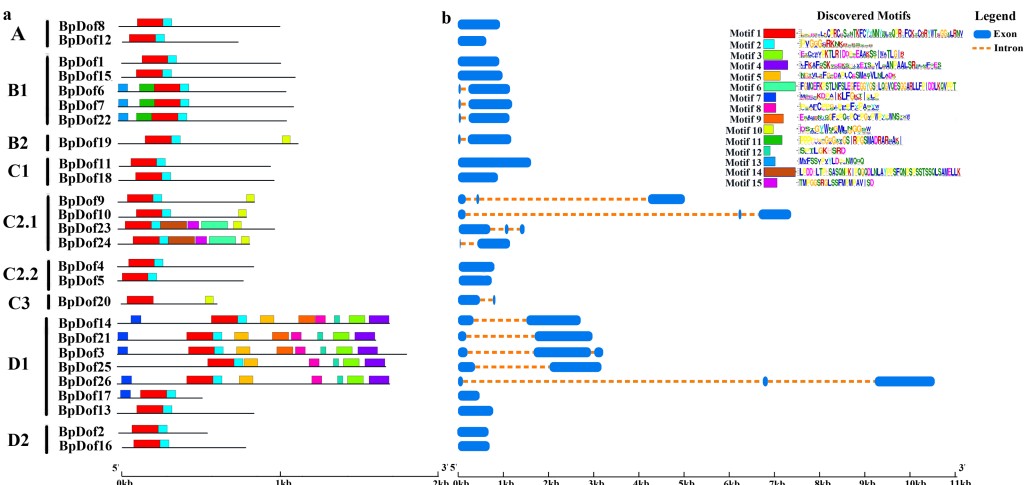

**Figure 3 The Motif and gene structure of the Dof protein family in birch.** (A) The distribution of motifs in BpDof proteins was examined. The 15 conserved motifs ($E$ value $< 10^{-5}$) in birch are represented with different color boxes, and the motif sequence logos are displayed in the upper right corner. The dark gray line shows the length of proteins. (B) Intron–exon patterns of 26 BpDof genes from birch. The legend was showed by the side of the motif sequence logos.

the 14 chromosomes of birch genome, with only one *BpDof* gene located on Chr02, Chr03, Chr10, and Chr13, respectively. Two *BpDof* genes were located on Chr04 and Chr08, respectively. Three *BpDof* genes were located on Chr07, Chr11, and Chr14, respectively. Four *BpDof* genes were located on Chr12, and five *BpDofs* on Chr06.

During plant evolution, gene duplication is a universal event in all organisms and is important in dissecting the novelties (*Lynch & Conery, 2000*). According to Fig. 4, seven duplication events were predicted among six chromosomes (Chr02, Chr04, Chr06, Chr08, Chr11, and Chr14), and these duplication events occurred in three subfamilies (B1, C2.1, and D1) of *BpDof* genes. All of the potential duplication events were inter-chromosomal duplication that occurred between two different chromosomes. Furthermore, the duplicated genes belonged to the same subfamilies, and the three groups of genes were found to have strong collinearity. *BpChr06G02126* and *BpChr02G19918* were in one group, *BpChr06G09621*, *BpChr11G09292*, and *BpChr14G12625* were in another group, and the last group included *BpChr04G00494*, *BpChr08G17028*, and *BpChr11G05806*. The Ka/Ks values of these *BpDof* genes were all less than 1 except those of *BpChr11G09292-BpChr14G12625* (Table 2), indicating that they have undergone strong purifying selection during the evolution. The Ks value of *BpChr11G09292-BpChr14G12625* was NaN leading to Ka/Ks value as NaN, indicating that gene duplication caused mutation at the nucleic acid level but not at the amino acid level.

## Analysis of promoter cis-elements of *BpDofs*

The putative cis-elements within the 2,000-bp genomic sequences upstream with the 5′-UTR of 26 *BpDofs* were predicted in the PlantCARE database. The results indicated that some cis-elements in 26 *BpDof* gene promoter regions were identified, including MYB and

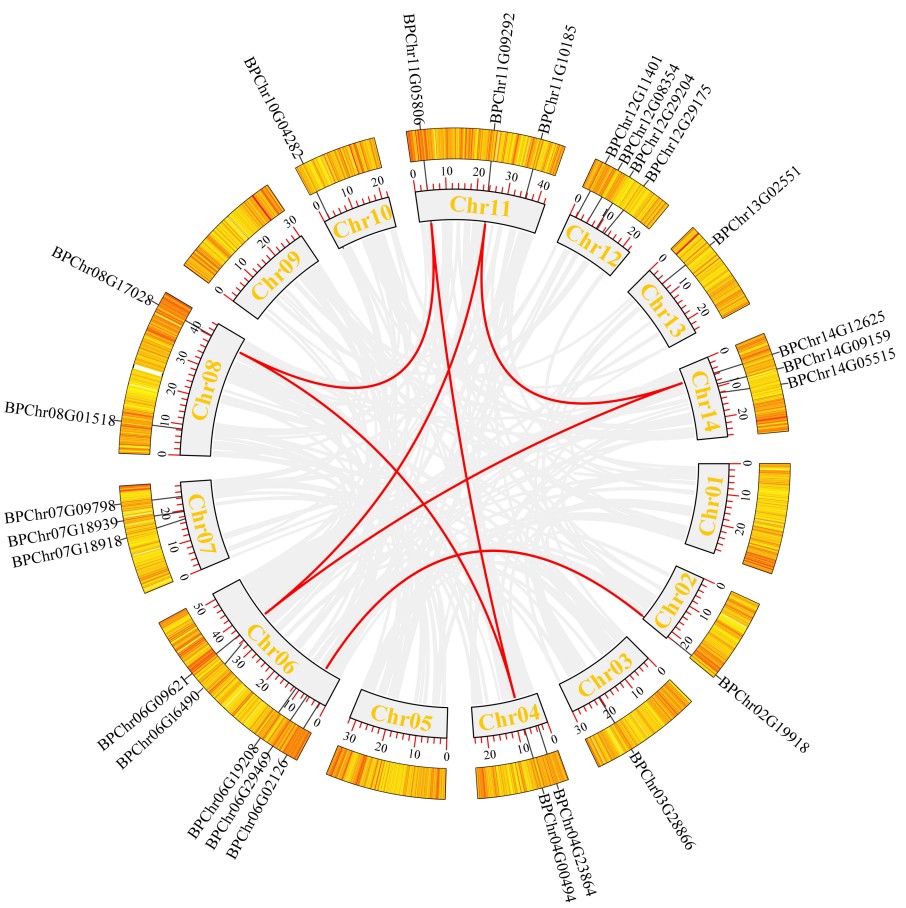

**Figure 4** **Chromosomal distribution and interchromosomal relationships of 26 *BpDof* genes.** Grey lines indicate all synteny blocks in the birch genome, and the red lines represent duplicated *BpDof* gene pairs. The thermal maps in the orange rectangles represent the gene density on the chromosomes. Grey rectangles represent birch chromosomes 01-14 and the chromosome number was displayed in the grey re-cetangles.

MBS (MYB-binding site, involved in drought inducibility), TC-rich repeats (defense and stress-responsive element), LTR (low temperature-responsive element), GT1-motif (light-responsive element), G-Box (light responsiveness element), GA-motif (light-responsive element), W-Box (WRKY-binding site, involved in abiotic stress responsiveness), ABRE (abscisic acid–responsive element), ERE, and GARE-motif (gibberellin-responsive element) (Fig. 5).

## Expression patterns of *BpDof* genes in response to drought stresses

The 2-month-old uniform seedlings were subjected to stress treatment to explore the change in birch *Dof* expression levels under drought stress. The relative expression levels of the 26 *BpDof* genes were analyzed in the roots, stems, and leaves of birch under 20% PEG6000 treatment compared with the control (water treatment) (Fig. 6 and Table S8).

In roots, the 26 *BpDof* genes were differentially expressed under PEG6000-simulated drought stress; most of them were upregulated at nearly all time points, and only a few
**Table 2  Duplication models for *BpDof* gene pairs in birch.**

| Duplicate gene pair | Ka | Ks | Ka/Ks | AverageS-sites | AverageN-sites |
|---|---|---|---|---|---|
| BPChr11G05806- BPChr04G00494 | 0.359251491 | 1.402780691 | 0.256099541 | 320.75 | 1077.25 |
| BPChr11G05806- BPChr08G17028 | 0.257338858 | 1.155021891 | 0.222799984 | 333.9166667 | 1112.083333 |
| BPChr04G00494- BPChr08G17028 | 0.303409975 | 1.775006855 | 0.170934537 | 310.0833333 | 1042.916667 |
| BPChr06G09621- BPChr11G09292 | 0.342798859 | 2.485336412 | 0.137928554 | 206.1666667 | 657.8333333 |
| BPChr06G09621- BPChr14G12625 | 0.318070599 | 1.25186786 | 0.254076815 | 220.4166667 | 697.5833333 |
| BPChr11G09292- BPChr14G12625 | 0.398755789 | NaN | NaN | 205.8333333 | 664.1666667 |
| BPChr06G02126- BPChr02G19918 | 0.683281032 | 3.032288451 | 0.225335104 | 152 | 475 |

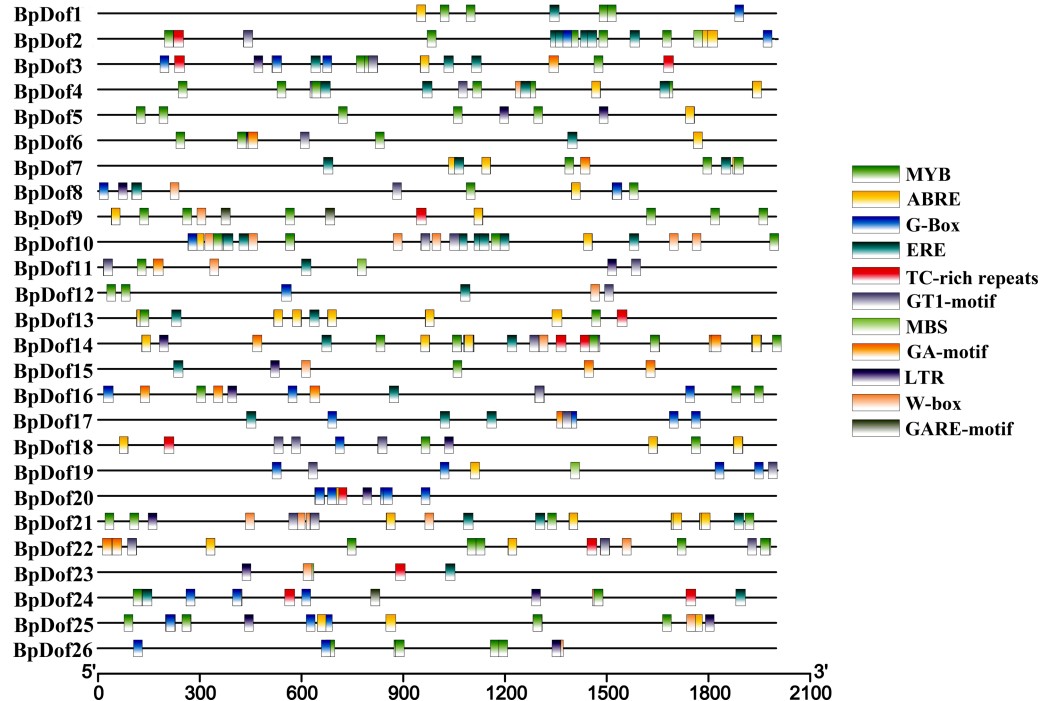

**Figure 5  Distribution of cis-elements in the promoters of 26 *BpDof* genes.** The 11 different cis-elements in the promoters of the 26 *BpDof* genes are represented in different color boxes.

genes at several time points were slightly downregulated. Compared with the control, the expression of *BpDof11* and *BpDof17* was significantly upregulated by about 4-fold to 199-fold, and about half of *BpDofs* reached their peak expression at 12 h after treatment. In stems, the expression of most *BpDofs* showed no obvious change. The expression of *BpDof3* (6-fold) at 12 h, *BpDof22* (7-fold) at 12 h, and *BpDof24* (7-fold) at 6 h was slightly upregulated. However, the expression of *BpDof5* was largely downregulated by about 26-fold at 12 h compared with the control. In leaves, the expression of most genes was significantly different compared with the control at most treatment time points. For example, the expression of *BpDof4* (322-fold), *BpDof5* (68-fold), and *BpDof14* (106-fold)

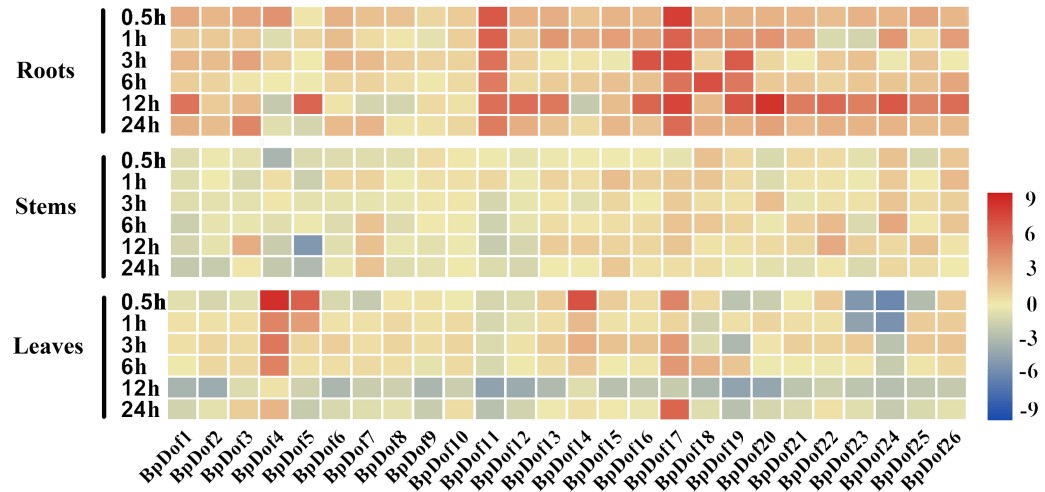

**Figure 6** Expression patterns of the 26 *BpDof* genes in the roots, stems, and leaves of 2-month-old *B. platyphylla* seedlings under drought stress (20% PEG6000) treatment. The gene expression of different tissues of birch plants was analyzed by qRT-PCR. The expression levels of the 26 *BpDof* genes after 0-h treatment was used as the control to detect the relative expression levels of the genes. All ratios are log2 transformed so that inductions and repressions of identical magnitude are numerically equal but opposite in sign. Log ratios of 0 (ratios of 1) are colored *yellow*, and increasingly positive (induction) or negative (repression) log ratios are *colored red* or *blue* with increasing intensity, respectively. *Red* means induction and *blue* repression in arrays.

peaked at 0.5 h, while that of *BpDof17* (59-fold) peaked at 24 h. However, the expression of *BpDof24* was downregulated at all treatment time points compared with the control.

## Plants overexpressing *BpDof4*, *BpDof11*, and *BpDof17* had reduced oxidative stress and cell membrane damage

The expression of *BpDof4, BpDof11*, and *BpDof17* was markedly upregulated at some times under drought treatment compared with the control. Therefore, these three genes were selected for further exploration. NBT and DAB staining was used to detect $O^{2-}$ and $H_2O_2$ levels so as to determine whether the drought tolerance was strengthened in the *BpDof4-, BpDof11-,* and *BpDof17* - overexpressing plants (Fig. 7). The leaves from *BpDof4, BpDof11, BpDof17*, WT and pROKII-35S plants were stained with NBT or DAB; the stained leaves from water-treated plants were used as controls. Under drought treatment, the $O^{2-}$ and $H_2O_2$ levels in the leaves of *BpDof4-, BpDof11-,* and *BpDof17-* overexpressing plants were greatly reduced compared with those in WT and pROKII-35S plants. The contents of $O^{2-}$ and $H_2O_2$ negatively reflected the reactive oxygen species (ROS) scavenging ability of plants. Therefore, the results indicated that *BpDof4-, BpDof11-,* and *BpDof17*-overexpressing plants had enhanced abilities to scavenge ROS. In addition, Evans blue staining was used to detect cell membrane damage. Further, *BpDof4-, BpDof11-,* and *BpDof17*-overexpressing plants showed less intense blue staining compared with WT and pROKII-35S plants under drought treatment, suggesting that they had decreased cell death.

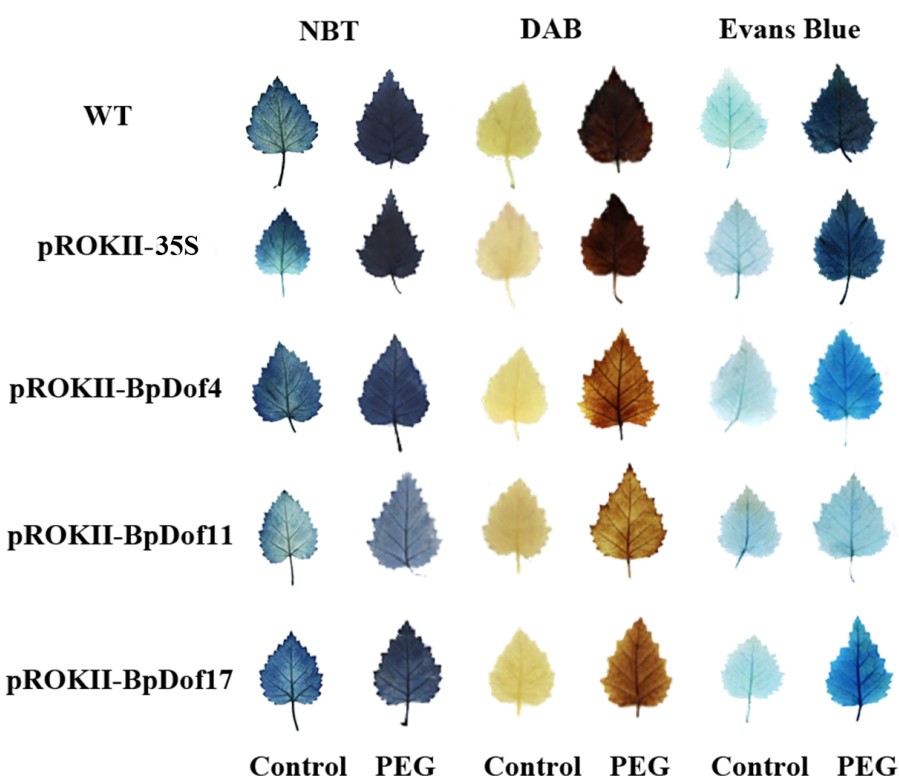

**Figure 7** **Analysis of ROS accumulation and cell membrane damage among transgenic and control birch plants.** Birch plants were transiently transformed with 35S: *BpDof4*, *BpDof11*, and *BpDof17* for overexpression, and empty pROKII was transiently transformed into plants as the control. After treatment with 20% PEG6000, the overexpression and control lines were individually stained with NBT to visualize $O^{2-}$, stained with DAB to visualize $H_2O_2$ level, and stained with Evans blue to visualize cell membrane damage.

## Physiological characterization of *BpDof4*, *BpDof11*, and *BpDof17*-overexpressing plants

In this study, SOD and POD activities, $H_2O_2$ content, and electrolyte leakage were used to assess the resistance of *BpDof4*-, *BpDof11*-, and *BpDof17* - overexpressing plants to drought stress, as well as that of the pROKII-35S transformants and WT birch plants (Fig. 8 and Table S9). Water treatment was used as control. SOD and POD play important roles in removing ROS in plants under stress. Results showed that the SOD activity in *BpDof11* - overexpressing plants was significantly higher than that in WT and pROKII-35S plants; however, the SOD activity in *BpDof4*- and *BpDof17*- overexpressing plants had not obvious change compared with WT and pROKII-35S plants. POD activity was significantly higher in *BpDof4*-, *BpDof11*-, and *BpDof17*- overexpressing plants compared with WT and pROKII-35S plants under drought stress. The $H_2O_2$ level decreased by 19.13%, 36.12%, and 7.01%, respectively, in *BpDof4*-, *BpDof11*-, and *BpDof17* - overexpressing plants compared with the control. Cell death was evaluated using the electrolyte leakage rate. Electrolyte leakage showed a slight decrease in *BpDof4*-, *BpDof11*-, and *BpDof17* - overexpressing plants compared with the pROKII-35S and WT plants under drought stress. These results

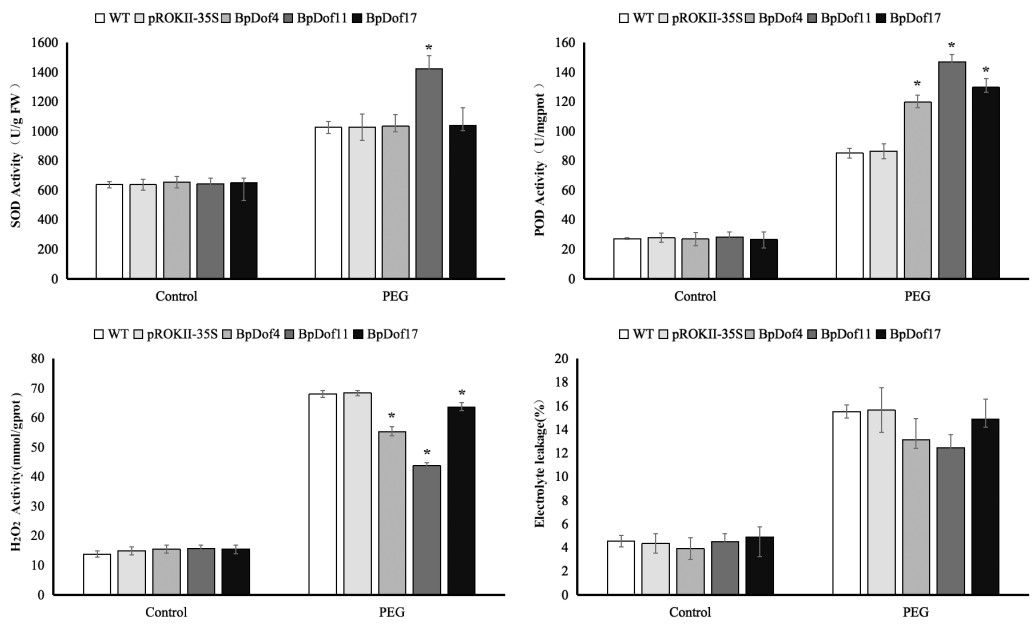

**Figure 8** **Physiological analysis of transgenic *BpDof4-*, *BpDof11-*, and *BpDof17* -overexpressing plants and control plants.** SOD and POD activities, $H_2O_2$ content, and electrolyte leakage in WT and transgenic plants under drought treatment conditions were measured. (A) Measurement of SOD activity. (B) Measurement of POD activity. (C) Measurement of $H_2O_2$ content. (D) Measurement of electrolyte leakage.

suggested that *BpDof4*, *BpDof11*, and *BpDof17* could enhance the ROS scavenging ability and inhibit cell death in plants.

## DISCUSSION

In this study, sequences of the 26 *BpDof* genes were obtained from the birch genome. All putative BpDof TFs had a single Dof domain with a C(x)2C(x)21C(x)2C zinc finger motif in the N-terminal region (Fig. 1), indicating that they were Dof proteins. The amino acid sequences in the C-terminal transcriptional regulatory domains of the 26 BpDof proteins varied, suggesting that the functions of the 26 BpDof proteins might be diverse (*Diaz et al., 2002*).

Based on the phylogenetic analysis of the *A. thaliana Dof* s (*Yanagisawa, 2002*; *Yu et al., 2020*), the birch *Dof* gene family was divided into four groups including nine subfamilies (group A-D, subfamily A, B1, B2, C1, C2.1, C2.2, C3, D1, and D2) (Fig. 2). Each of the birch *Dof* genes had one or more homologous genes in *Arabidopsis*, implying that *Dof* genes in birch might play similar roles as their homologs in *Arabidopsis* (*Zhou et al., 2020*). Motifs 1 and 2 were commonly shared by most *BpDof* family members (Fig. 3A), which was consistent with the results obtained from *Arabidopsis*, rice, cucumber, and tomato (*Cai et al., 2013*; *Lijavetzky, Carbonero & Vicente-Carbajosa, 2003*; *Wen et al., 2016*), thus suggesting that BpDof TFs were evolutionarily conserved in plants. The exon–intron structure of genes can provide insights into the evolutionary relationships within certain gene families (*Zhou et al., 2020*; *Zhou et al., 2018*). In the present study, the numbers of
introns in *BpDof* genes ranged from 0 to a maximum of 2, and most *BpDof* genes contained a single intron or no intron at all (Fig. 3B). Similar results have been reported in many other plant species, such as *Arabidopsis* (*Kushwaha et al., 2011*), rice (*Lijavetzky, Carbonero & Vicente-Carbajosa, 2003*), cucumber (*Wen et al., 2016*), poplar (*Wang et al., 2017*), eggplant (*Wei et al., 2018*), and pear (*Liu et al., 2020*), indicating that the exon–intron structure of *Dof* genes is highly conserved across plant species, which may be related to their similar functions. In a specific gene family, the integration and realignment of gene fragments might result in exon–intron variation (*Xu et al., 2016a*), and the disparate exon–intron structures of the *BpDof* genes indicated that they may play different roles.

The putative cis-elements in promoters of 26 *BpDofs* were analyzed in the PlantCARE database. The results (Fig. 5) showed that MYB, MBS, TC-rich, LTR, GT1-motif, G-Box, GA-motif, W-Box, ABRE, ERE, and GARE-motifs, which were related to drought tolerance (*Zhang et al., 2019*), flavonoid biosynthesis (*Wang et al., 2018*), defense and stress responsiveness (*Zhang et al., 2005*), low temperature tolerance (*Feng et al., 2019*), light response (*Zhang et al., 2013*; *Zhu et al., 2015*), abiotic stress responsiveness (*Yan et al., 2019*), abscisic acid responsiveness (*Choi et al., 2005*), ethylene responsiveness (*Rawat et al., 2005*), and gibberellin responsiveness, were found in promoters of the 26 *BpDof* genes. Among these, MYB, MBS, G-Box, GT1-motif, ABRE, and ERE cis-elements were abundant, thus suggesting that these *BpDof* genes might be involved in drought tolerance, light response, and ABA- and ethylene-responsive signaling.

Under adverse environmental conditions, plants have developed many strategies in response to various abiotic stresses (*Ma et al., 2015*). Previous studies have indicated that some *Dof* genes might play essential roles in response to abiotic stress (*Gu et al., 2019*; *Zhao et al., 2019*). Furthermore, overexpression of *Dof*s significantly increased the salinity and drought tolerance in transgenic plants (*Cheng et al., 2018*; *He et al., 2015*; *Liu et al., 2019*). In this study, the expression of most *BpDof* genes was significantly different among birch roots, stems, and leaves under PEG6000 treatment from 0.5 to 24 h (Fig. 6), which suggested that these *BpDof* genes could be regulated by drought stress and might play key roles in response to drought stress. Under drought treatment, the expression levels of most *BpDof* genes were up-regulated in roots; no obvious expression differences were observed in stems, but the expression of most *BpDof* genes was downregulated in leaves (Fig. 6), suggesting *Dof* genes were differentially expressed in different birch tissues as reported in other plant species (*Gupta et al., 2018*; *Song et al., 2016*). Meanwhile, our results also indicated that the responses of most *BpDof* genes to drought stress were tissue-specific in birch. The differential expression of the 26 *BpDof* genes under different treatment time also suggested that the signaling pathways in plant response to drought stress were complex.

Plants are commonly exposed to various adverse situations, which cause the accumulation of ROS (*Wang et al., 2005*). Therefore, ROS scavenging is important for plant resistance to various stresses. Two major ROS species $O^{2-}$ and $H_2O_2$ are important molecules in cells, which are involved in oxidative injuries and stress signaling (*Zhang et al., 2011*). In this study, NBT and DAB staining showed that the ROS accumulation was lower in *BpDof4-*, *BpDof11-*, and *BpDof17-* overexpressing plants than in WT plants under drought treatment (Fig. 7), which was consistent with the levels of $H_2O_2$ (Fig. 8C). The

SOD and POD played vital roles in removing ROS. The SOD and POD activities were significantly higher in *BpDof4-, BpDof11-,* and *BpDof17* - overexpressing plants than in WT plants under drought stress (Figs. 8A and 8B). These results showed that *BpDof4, BpDof11,* and *BpDof17* could enhance the ROS scavenging ability by improving the SOD and POD activities. The result of Evans blue staining (Fig. 7) was consistent with electrolyte leakage rates (Fig. 8D). The *BpDof4-, BpDof11-,* and *BpDof17* - overexpressing plants had significantly less intense Evans blue staining and lower electrolyte leakage rates compared with WT plants. The results indicated that *BpDof4, BpDof11,* and *BpDof17* reduced the cell death to enhance the resistance to stress in plants.

## CONCLUSIONS

The comprehensive analysis of Dof transpcrition factors was performed in the genome of birch. A total of 26 *BpDof* genes encoding Dof transpcrition factors were identified from birch, which were classified into four groups and nine subgroups: A, B1, B2, C1, C2.1, C2.2, C3, D1 and D2. The gene structure, conserved motifs and phylogenetic relationships of 26 Dof genes were analyzed. Almost all of the BpDof proteins contained motif1 and motif2 which were considered as the conserved Dof domains. We also investigated the expression patterns of *Dof* genes at roots, stems and leaves of birch under drought treatment. Moreover, the resistance of *BpDof4, BpDof11* and *BpDof17* to drought stress in transient transgenic birch plants was conducted. Our results provide valuable information for further understanding of the regulatory mechanisms of BpDof transcription factors in response to abiotic stress.

### Funding

This work was supported by the Opening Project of State Key Laboratory of Tree Genetics and Breeding (K2017201) and the National Natural Science Foundation of China (31700587). The funders had no role in study design, data collection and analysis, decision to publish, or preparation of the manuscript.

### Grant Disclosures

The following grant information was disclosed by the authors:
Opening Project of State Key Laboratory of Tree Genetics and Breeding: K2017201.
National Natural Science Foundation of China: 31700587.

### Competing Interests

The authors declare there are no competing interests.

### Author Contributions

- Shilin Sun performed the experiments, analyzed the data, prepared figures and/or tables, authored or reviewed drafts of the paper, and approved the final draft.
- Bo Wang performed the experiments, prepared figures and/or tables, authored or reviewed drafts of the paper, and approved the final draft.

- Qi Jiang performed the experiments, authored or reviewed drafts of the paper, and approved the final draft.
- Zhuoran Li and Site Jia analyzed the data, prepared figures and/or tables, and approved the final draft.
- Yucheng Wang conceived and designed the experiments, authored or reviewed drafts of the paper, and approved the final draft.
- Huiyan Guo conceived and designed the experiments, prepared figures and/or tables, authored or reviewed drafts of the paper, and approved the final draft.

## Data Availability

The raw data are available in the Supplemental Files and GenBank: MW538484 to MW538509.

## Supplemental Information

Supplemental information for this article can be found online at http://dx.doi.org/10.7717/peerj.11938#supplemental-information.

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
