# Peer review of "Genome-wide analysis of BpDof genes and the tolerance to drought stress in birch (Betula platyphylla)"

_PeerJ, doi:10.7717/peerj.11938_

## Round 0.1 · original submission · Major Revisions

Two reviewers have different views on your manuscript. I would like to provide you an opportunity to revise your manuscript based on the comments from reviewers.

Reviewer 1 ·

Basic reporting

A few sentences in this manuscript are ambiguous.

Experimental design

The author needs to describe the methods of gene screening in detail.

Validity of the findings

no comment.

Additional comments

In this study, the authors identified 26 Dof family genes from birch tree. Then the phylogenetic relationship, gene structure, conserved motifs of these BpDofs were analyzed. Expression analysis showed that several BpDofs response to PEG treatment significantly. Three BpDof genes with the most significant expression changes under PEG treatment were selected for subsequent transgene using a transient expression method. Determination of stress-related indexes showed that these genes could enhance the drought tolerance of transgenic plants. In general this manuscript is interesting and provides valuable information. However, the manuscript was poorly written and had many errors. The results section is oversimplified in some places. The discussion section is logically confusing and needs to be rewritten, and some of the sentences in this section are duplicated with the results section. Part of the bioinformatics analysis results and experimental results need to be further confirmed.

Keywords:
“Abiotic stress”, change to “Drought stress”.

Introduction:
The third paragraph (lines 61-85) in this section is too long. I think this paragraph can be divide the into two parts at line 80. In addition, the fourth paragraph (line86-89) should be merged into the preceding paragraph.

Materials & Methods:
Line 106, how were the 26 BpDofs genes screened? The author needs to describe the methods of gene screening in detail.

In the title of Table S1, The letter “BpDof” should not be italicized when described as a transcription factor. Letters are italic only if they are the name of a gene. There are many of the same mistakes in this manuscript.

Line 107-111, this sentence is ambiguous and needs to be rewritten. In addition, if the author want to identifiy the conserved domains contained in these BpDof proteins, they should not directly use the Dof domain file (Pfam PF02701) for identification. They do not know whether these DOF proteins contain other conserved domains.

Line 121-123 (Table S2), 39 AtDof genes are listed in this Table. However, only 26 BpDof genes were obtained in this study. This makes it doubtful whether there are many omissions in the screening results of this study.

Line 143, the method of PEG treatment needs to be described in detail. For example, how much PEG solution is poured in this treatment?

Line 175, the author states that the “pROKII-35S was used as a control”. However, in the results section (line 273; Figure 7), the authors used WT plants as controls

Results:
Line 189-190, the authors should briefly describe how they screened for Dof genes in birch.

Line 196-198, there are no charts or supplementary materials in the paper to show the results described in this sentence.

Line 199, as a genome-wide analysis of Dof family genes, collinearity analysis should be performed in this study.

Line 201, two AtDof genes were randomly selected for sequence alignment, which is not as good as selecting representative genes in this family that have been studied extensively.

Line 206-209 (Figure 2), based on the phylogenetic tree analysis, the authors divided BpDof proteins into five groups, which were named II to VI. However, in Figure3 (line 215), the authors divided these Dof proteins into 6 groups, and the naming method was changed to A-F. The author needs to further verify the grouping results and unify the naming method. In addition, 38 AtDofs protein sequences were used in Figure 2, while 39 Atdofs protein sequences were listed in Table S2. Please check whether the figure legend of Figure 2 is incorrect or indicate which sequence of AtDofs is not used for the construction of the phylogenetic tree.

I suggest combining Fig4 and Fig3 into one figure.

Line 262-263 and line 266, “BpDof417、BpDof424、BpDof411” ? Please check.

Line 263-264, Dof4 and 17 are up-regulated, while Dof24 is down-regulated. The description should be clear. In addition, the expression of Dof14 in leaves also changed greatly (Figure 6), the expression level even higher than that of Dof17, which should not be omitted here.

Line 269, the methods of transgene and PEG treatment should be briefly described here. In particular, the author should make it clear that the transient expression method is used in this study.

Line 286, the authors said that the SOD activity of the transgenic material was significantly higher than that of WT, but there was no difference between the enzyme activity of two transgenic lines (Dof4, Dof17) and WT (Figure 8).

Line 291, please check whether there is a significant difference between Dof17-overexpressing plants and WT in Figure 8D.

Discussion:
Line 302-304, the AtDofs were divided into 7 groups in other studies. Why are there fewer groups in this study when all the sequences of AtDofs are used for phylogenetic tree analysis? Further verification and discussion are needed.

Line 304-305, delete this sentence or put it in another appropriate place.

Line 306, merge this paragraph with the previous one.

Line 314, delete this sentence or move it to the results section.

Line315, this part needs to be further discussed in combination with the similarity of gene structure within the same group and the differences between different groups.

Line335-336, further discussion should be conducted on the differences in expression patterns of the same gene in different tissues (roots, leaves), such as whether the response and regulatory functions of some Dof genes under drought stress are tissue-specific.

Conclusions:
Line 353, “molecular analysis” is ambiguous.

Line 356, change “BpDof genes” to “BpDof proteins”

Line 361, showed “enhanced” ROS scavenging capability

Reviewer 2 ·

Basic reporting

no comment

Experimental design

no comment

Validity of the findings

no comment

Additional comments

The manuscript by Sun et al. reports Genome-wide analysis of BpDof genes and the tolerance to drought stress in birch. The author analyzed the characterization and expression of the BpDof genes and the tolerance to drought stress, indicating that some of them can enhance the resistance to drought stress. I think this work is interesting, but, there are several issues need be revised.
1. Please check the full text. The name of gene is in italics and the name of protein is in normal.
2. Please keep the format consistency of all figures.
3. Line 149: What size of the birch seedlings was treated with 20% PEG6000?
4. Line 174: Please add the restriction endonuclease used in the vector construction.
5. Line 184 and 185: Please add the dissolving solutions of DAB and Evans blue.

---

## Round 0.2 · Minor Revisions

Please revise your manuscript according to the comments from the reviewer.

Reviewer 3 ·

Basic reporting

Although this is the first round for me to assess the academic quality of the manuscript submitted by Sun et. al., I can find the manuscript has been extensively revised according to the tracking changes. The current (revised) version of the manuscript is much improved. Dofs are plant specific TFs that play various roles in developmental processes and environmental stresses. Genome wide identification and analysis of Dof genes is useful to further reveal their molecular functions. The manuscript is overall well designed and organized. However, some minor reversions are needed before publication.

Experimental design

The manuscript is well designed, I have no special comments to this part.

Validity of the findings

The results and findings obtained in this study is meaningful for researchers who are interested in this field.

Additional comments

1. For the method section, the authors should refer to the literatures of the softwares, such as BLASTX (L111), MEGA (L127) .
2. L233 Populus trichocarpa, not Populus trichocarp. In addition, please the genome version used in this analysis.
3. For the conclusion section. please do not simply repeat the results again.

---

## Round 0.3 · Minor Revisions

Please define the scale for Figure 6 (heatmap).

Reviewer 2 ·

Basic reporting

no comment

Experimental design

no comment

Validity of the findings

no comment

Additional comments

Based on my suggestion in the first review, the authors have addressed my concerns appropriately. The authors did a commendable job in putting together all the Betula platyphylla available Dof genes resources and performing in-silico analysis, this work is meaningful. So, the manuscript can be accepted for publication in PeerJ.

Reviewer 3 ·

Basic reporting

The authors have addressed all my concerns and the revised manuscript is much improved now. I have no further comments on the manuscript.

Experimental design

I have no special comments on the experimental design part.

Validity of the findings

No comment.

Additional comments

Thanks for your quick response, I have no other comments now.

---

## Round 0.4 · accepted · Accept

Your revised version of the manuscript is accepted.